# "I think there is a certain worm that disturbs the head": Community perspectives and care pathways for psychosis in Malawi

Dennis Chasweka[1]*, Kate Chidzalo[1], Wakumanya Sibande[1], Demoubly Kokota[1,2], Michael Udedi[3], Martyn Pickersgill[4], Charlotte Hanlon[2,5], Stephen M. Lawrie[2], Lucinda Manda-Taylor[1], Sumeet Jain[6]

1 Department of Bioethics, Behavioral and Health Social Sciences, Kamuzu University of Health Sciences, Blantyre, Malawi, 2 Division of Psychiatry, Centre for Clinical Brain Sciences, University of Edinburgh, Edinburgh, United Kingdom, 3 Curative and Medical Rehabilitation Services Directorate, Ministry of Health, Lilongwe, Malawi, 4 Centre for Biomedicine, Self and Society, Usher Institute, College of Medicine and Veterinary Medicine, University of Edinburgh. Edinburgh, United Kingdom, 5 Department of Psychiatry, School of Medicine, College of Health Sciences, Addis Ababa University, Addis Ababa, Ethiopia, 6 Social Work Subject Area, School of Social and Political Science, University of Edinburgh, Scotland, United Kingdom

* dchasweka@medcol.mw

## Abstract

Psychosis remains a significant global health challenge, with people in low- and middle-income countries (LMICs) like Malawi disproportionately impacted. Yet, little is known about community perspectives, treatment-seeking practices, and acceptable pathways for psychosis management. This study explored diverse community perspectives of psychosis to inform the development of a community-based psychosis detection and management system. This exploratory qualitative study was conducted in Malawi's Salima and Chiradzulu districts between October and December 2023. Seventy-six participants, including traditional healers, religious leaders, caregivers, and PWLE, were purposively sampled for 16 in-depth interviews (IDIs) and six focus group discussions (FGDs). Data were collected using semi-structured interviews, transcribed, and thematically analysed. The majority across all participants largely attributed psychosis to witchcraft, smoking cannabis, and alcohol use. Traditional healers and religious leaders were commonly the first points of care, and preferred over biomedical services. Participants recommended active community involvement, engagement of health surveillance assistants (HSAs), stronger partnerships between community leaders and health workers, and resource availability for an ideal community-based psychosis detection and management system. Cultural norms and practices strongly influence psychosis care pathways in Malawi, in the context of severe economic challenges which shape the provision of healthcare. Future interventions should prioritise culturally sensitive community outreach programs to improve psychosis management.

**Data availability statement:** The data supporting this study are available upon reasonable request through a Data Sharing Agreement with the Research and Ethics Committees of Kamuzu University of Health Sciences (KUHeS) and the University of Edinburgh. Interested parties may request access by emailing comrecassadmin@kuhes.ac.mw, including "PROMISE Community Stakeholders Paper Data Sharing" and "PI: Associate Professor Lucinda Manda-Taylor" in the subject line and body of the email.

**Funding:** Funding for this study was provided by Wellcome Trust(www.wellcome.ot) grant number 223615/Z21/Z to Professor Stephen M. Lawrie as the Principal Investigator & CH, SJ, LMT, DK, WS, DC, KC, MP, MU as co-investigators. The funders had no role in study design, data collection and analysis, decision to publish, or preparation of the manuscript.

**Competing interests:** The authors have declared that no competing interests exist.

## Introduction

Psychosis remains a significant global health challenge. According to the Global Burden of Disease study (2019), psychosis has a greater disease burden than more prevalent mental disorders such as depression and anxiety [1]. Early treatment of psychosis is crucial for better outcomes, as a longer duration of untreated psychosis (DUP) is associated with more severe symptoms, diminished social functioning, and increased caregiver responsibilities Evidence suggests that DUP tends to be longer in low- and middle-income countries (LMICs) compared to high-income countries (HICs) [2]. For instance, studies conducted in Malawi report delays of 42–71 months before individuals seek biomedical care after the first episode of psychosis [3,4]. While the prevalence of psychosis in many sub-Saharan African countries like Malawi is currently unknown, risk factors such as poverty, a large youth population, malaria, and cannabis use suggest that it could be relatively high [5].

One key factor influencing DUP is treatment-seeking pathways. Research shows that nearly half of all people with mental disorders in Africa seek treatment from traditional healers or religious leaders before considering biomedical care [6]. These healers play a central role in mental healthcare, as communities widely perceive traditional and faith healing practices as effective [7]. A recent systematic review on barriers to mental health treatment in Africa identified key challenges, including community attitudinal barriers such as stigma, infrastructural barriers resulting from a lack of healthcare facilities and trained professionals, and economic barriers linked to poverty [8]. For instance, in Malawi, specialist biomedical care is minimal, with only four tertiary referral hospitals staffed by four psychiatrists and three clinical psychologists serving the entire population of 20 million people [5]. An absence of specialist care contributes to a reliance on non-biomedical pathways, such that engagement with formal healthcare services can be delayed - contributing to longer DUP and ultimately poorer outcomes for individuals with psychosis [6].

Addressing DUP requires effective and accessible treatment strategies once individuals enter the formal healthcare system. In order to better serve individuals and their families affected by psychosis, there is a need to develop community-based approaches to early identification and care for people with psychosis This proposition underpins the Wellcome-funded PROMISE study [5], described more fully below, whose study team members comprise the authors of this paper. However, there is currently limited evidence within the literature of how people with lived experience (PWLE) and others in their community might regard such a system. Indeed, there is a strikingly paucity of literature on perspectives around community beliefs, knowledge, pathways to care, and experiences related to treatment-seeking and management of psychosis in Malawi. As such, this paper explores the perspectives of traditional healers, religious leaders, caregivers, and people with lived experience (PWLE) of psychosis in Malawi, with a view to informing the development of a community-based psychosis detection and management system. The following sub-sections highlight some of the existing literature on influence of traditional healers and religious leaders on community-based psychosis systems, experiences and challenges faced by PWLE and caregivers in management of psychosis in Malawi.

## The role of traditional healers and religious leaders in community-based mental health care

Traditional healers and religious leaders play a popular and integral role in the mental health treatment-seeking pathway across Africa, including Malawi [6,9,10]. In many communities, they are often the first point of contact for individuals experiencing mental illness. For example, a study conducted at Zomba Mental Hospital found that approximately 23% of newly admitted psychiatric patients initially sought treatment from traditional healers [11]. This trend is not limited to mental illness; for instance, at Queen Elizabeth Central Hospital, a tertiary facility in Blantyre, Malawi, 43% of general medical inpatients first sought consultation with traditional healers before visiting the hospital [12].

Traditional healers interpret mental illness through spiritual and other frameworks, often attributing symptoms to causes such as ancestral spirits, witchcraft, or curses. Their treatment methods include a combination of herbal remedies, spiritual rituals, and guidance to restore the balance between the individual and the community [13,14]. Religious leaders are similarly influential, offering prayer, guidance, and healing sermons, as many communities associate mental illness with possession by evil spirits.

The popularity of traditional healers and religious leaders, particularly in rural areas, stems from their availability, accessibility, and shared values. Their explanatory models resonate strongly within these communities [15]. Despite their prominence, there is a paucity of evidence regarding collaboration or integration between traditional healing practices and biomedical approaches in Malawi. Both traditional healers and biomedical health practitioners have displayed mixed reactions to collaboration initiatives, like cross-referrals, with biomedical practitioners expressing reluctance towards such collaboration and integration [16,17]. However, there is a long history of such collaborations in various parts of the African continent and an emerging literature on processes of developing successful collaborations [18,19]. Exploring such synergies could provide opportunities to address gaps in mental health service provision and reduce barriers to timely care.

## Experiences and Challenges of People with Lived Experience (PWLE) of Mental Ill-Health and Caregivers

Family members and other caregivers also play a critical role in supporting individuals with mental ill-health in their homes and communities. Family members and caregivers typically manage responsibilities such as symptom monitoring, medication management, and providing emotional support - often with limited resources or formal training [20]. However, caregiving can come with significant challenges, including emotional distress, disruption of family functioning, exposure to physical violence, financial strain, and social stigma [21,22]. In the face of these challenges, caregivers draw on various strategies, including the religious or spiritual practices highlighted above [23,24]. Despite their critical role, caregivers are often excluded from mental health interventions and policy development, and their needs - such as better health information and support systems - remain under-addressed [25].

PWLE of mental illness also face significant challenges in accessing healthcare and navigating daily life within their communities. Common issues include stigma, discrimination, social exclusion, financial difficulties, and limited access to healthcare and medications [26]. Although there has been strong advocacy for involving PWLE in mental health systems, their participation remains minimal in most African settings [27]. PWLE frequently express the need for greater inclusion in work opportunities, access to comprehensive health information, and involvement in decision-making processes to address their unmet needs. Engaging PWLE in designing interventions and policies could provide valuable insights into strengthening mental health systems and improving their quality of life. However, evidence of their involvement in these processes remains scarce, highlighting a critical gap in current mental health practices.

## The PROMISE Project

The Psychosis Recovery Orientation in Malawi through Improved Services and Engagement (PROMISE) study is a five-year research project funded by the Wellcome Trust. The primary objective of this project is to enhance existing mental

health services and programs by developing sustainable, socially and culturally acceptable psychosis detection and management systems. In so doing, the project seeks to improve outcomes for people with psychosis in Malawi [5].

This paper reports on data collected during Work-Package 1 of PROMISE, which explored the perspectives and experiences of various community-based stakeholders—traditional and religious healers, PWLE, and caregivers—to inform the development of psychosis detection and management intervention by Health Surveillance Assistants (HSAs), community-based health workers. Specifically, this paper addresses the following questions:

1. How is psychosis experienced and perceived in the community?

2. What beliefs and knowledge are leveraged to understand psychosis?

3. How do community perceptions and experiences relate to treatment-seeking and management, priorities, and preferences for care?

4. What are the current challenges, and what would represent an ideal system and desirable outcomes among PWLE and their families?

## Methods

### Ethics statement

The Kamuzu University of Health Sciences Ethical Committee (P.03/23/4034) and Edinburgh Medical School Research Ethics Committee (22-EMREC-044) approved the PROMISE study (P.03/23/4034). The ethical committees of Salima and Chiradzulu district hospitals also approved it. All participants provided informed consent before participating in the research, ensuring their voluntary involvement and understanding of its purpose and procedures. Ethical approval was granted in March 2023 prior to any data collection.

### Study Setting

The study was conducted in two districts of Malawi: Salima, a lakeshore district in the central region, and Chiradzulu, located in the southern region. Salima is 122 km away from Lilongwe, Malawi's capital city, and covers an area of 2,196 km$^2$. The population of Salima District was approximately 551,022 as of 2023 [28]. Chiradzulu is located about 27 km from Blantyre, Malawi's commercial city. It covers an area of 761 km$^2$ and has a population of approximately 360,000 (NSO, 2018) [29].

### Study design and Population

This exploratory qualitative study was conducted between October and December 2023. The study population consisted of traditional healers, religious leaders, PWLE, and caregivers.

### Sampling and Recruitment

Focus group discussions (FGDs) and in-depth interviews (IDIs) were conducted with key informants: traditional healers, religious leaders, PWLE, and caregivers. Purposive sampling was used to select the participants. The sampling method was chosen to accommodate diverse experiences and perspectives. Participants were intentionally selected for focus groups and interviews within the study area based on age, gender, profession, geographical location and experience of psychosis. This purposive sampling strategy is a widely utilised qualitative research technique for identifying and selecting cases that provide in-depth, information-rich insights [30]. Geographic location was also considered to ensure a diverse range of participants' experiences and perspectives from semi-urban and rural areas. Inclusion criteria for PWLE participants required a confirmed diagnosis of psychosis and assessment as being able

to participate in the discussions. Participant recruitment was facilitated through consultations and support from district mental health teams and peer support groups. This included the patient advocacy group, the Malawi Mental Health Service Users and their Caregivers' Association (MeHUCA), some of whose staff were also part of our wider study team.

Eight IDIs were conducted in each district: two with PWLE, two with caregivers, two with traditional healers, and two with religious leaders. Additionally, two FGDs were conducted in each district: one with PWLE and their caregivers, and another with traditional healers and religious leaders. Only individuals aged 18 years and over who were deemed able to provide informed consent were included in the study. Participants were assigned numbers to ensure confidentiality and anonymity; no personal identifiers were included during data collection.

A semi-structured interview guide was used to facilitate IDI and FGD discussions. The guide prompted discussion about the participants' perspectives about psychosis, community perceptions of psychosis, the potential role of HSAs in psychosis detection and management, and what an ideal community-based psychosis detection and management system would look like. This data collection tool was piloted to ensure clarity and consistency: one FGD and four IDIs (with a PWLE, caregiver, traditional healer, and religious leader) were conducted. This informed some adjustments to wording and phrasing, which were corrected in the final guide.

There is no specific local term for psychosis in Malawi. To avoid confusion with other mental disorders in responses, particularly epilepsy, which is commonly known as *khunyu* in Chichewa, psychosis was described using the term *misala*, meaning "madness." This term is widely used to refer to individuals with psychosis and is often associated with non-normative behaviours such as talking to oneself or wandering naked in the streets [10].

Within qualitative research, data saturation occurs when no new themes, concepts, or insights emerge from additional data collection. Hennink and colleagues emphasise that the depth and richness of the data, rather than the number of participants, are critical determinants of saturation. This is particularly relevant in studies with well-defined research questions and homogeneous participant groups, where redundancy in information is more likely to be achieved with smaller sample sizes [31]. Within our research, no new themes emerged after 16 IDIs and four FGDs, implying data saturation.

## Data collection and analysis

Three PROMISE researchers (DC, KC, and WS) conducted the data collection. IDIs were conducted at the participants' nearest health centres. A private space was provided for the interviews. FGDs were convened at mutually agreed-upon conference centres near the district hospitals, providing participants with a neutral and comfortable environment.

All interviews were conducted in Chichewa, Malawi's widely spoken national language. Interviews were audio-recorded with the participant's consent, and notes were taken to capture critical points. The recordings were transcribed verbatim and then translated directly into English by the research assistants (DC, KC, and WS).

Data analysis employed thematic analysis, which involves systematically identifying, organising, and interpreting meaningful patterns within the data. This process included data familiarisation through repeated reading of transcripts to enhance understanding. Four researchers (LMT, WS, DC, and KC) independently coded the transcripts to identify patterns and preliminary themes emerging from the data. Initial coding of the interview transcripts, conducted inductively and deductively, helped identify preliminary parent and child codes. Any discrepancies among the coders were addressed through merging and adding codes, ultimately producing a unified codebook. Subsequently, an independent coder, SJ, reviewed the same transcripts to align with the established codes. DC, KC, and WS then coded the remaining transcripts in Nvivo14 software, organising them into themes and sub-themes. Following re-organisation, comparison, and categorisation, overlapping themes were consolidated for further analysis, interpretation, and reporting. Key quotes were selected from the data to exemplify significant findings.

### Researchers' positionality

The researchers included social scientists and mental health practitioners and were reflexive about how this positionality could influence interpretations of participants' perspectives and experiences. This included journaling and team discussions throughout the study to examine individual and collective interpretations and maintain analytical rigour. Such reflexivity was particularly important given the marginalisation of PWLE of psychosis, and the structural discrimination and interpersonal stigma which they experience. It was important to the study team to try hard to ensure that this research did not inadvertently reproduce some of those challenges to our participants. All involved in data collection and coding had prior experience conducting qualitative research with caregivers, PWLE, religious leaders, and traditional healers. This background provided valuable contextual insight and facilitated rapport-building with participants.

### Credibility and trustworthiness

The study research assistants underwent training in qualitative data collection and analysis, contributing to methodological rigour. During IDIs, detailed notes and key points were summarised and clarified with participants to verify accuracy before the session concluded. Data quality checks were conducted during transcription by assigning a different researcher to transcribe a sample of interviews and comparing the results for consistency. Triangulation, incorporating field notes and cross-validation of data, further enhanced credibility. As noted, the coding process involved four researchers (DC, KC, WS, and LMT) independently presenting their parent and child codes, which were then merged into a unified codebook reflecting standard codes agreed upon by the team. All procedures were meticulously documented throughout data collection and analysis to ensure dependability and transparency, enabling future study replication.

## Results

A total of 16 participants (8 males and 8 females) took part in the IDIs, with equal representation from Chiradzulu (8) and Salima (8) and an equal gender distribution across these groups. Additionally, 60 participants (12 PWLE, 13 carers, 13 religious leaders, and 12 traditional healers) participated in the six FGDs. Of these, two-thirds (70.8%) of the participants were male. The median age of all IDI and FGD participants was 39 years (18–60). See Table 1.

The study findings are presented below, organised around four themes that reflect the focus of the study: characterisation of psychosis, community treatment-seeking approaches, challenges with care, and an ideal community-based psychosis detection and management system. A summary of key themes is provided in Table 2, above.

### Characterising psychosis

#### Description and associated symptoms

The participants provided various descriptions of individuals with psychosis. Most described people with psychosis based on observable behaviours, speech, and patterns or habits that were seen as deviating from community norms. Commonly mentioned attributes or propensities included running away from home, poor personal hygiene, wandering, physically attacking others, and collecting trash. One participant said:

> *"The person will have changed their speech, behaviour and even dressing, their clothes are filthy, and they do not even care about themselves."* (Male traditional healer, IDI-CZ)

Another reflected:

> *"They do not have time to take care of themselves; they just wander around, and they do not know where they are going and their destination."* (Male PWLE, IDI-CZ)

**Table 1. Demographic characteristics of interview participants.**

| Participant Demographic Characteristics | |
|---|---|
| **In-depth Interviews (n = 16)** | |
| **Characteristics** | **n** |
| **Sex** | |
| Male | 8 |
| Female | 8 |
| **Education Level** | |
| Primary | 7 |
| Secondary | 8 |
| Tertiary | 0 |
| Other | 1 |
| **Occupation** | |
| Religious leader (Christian) | 3 |
| Religious leader (Muslim) | 1 |
| Traditional healer | 4 |
| Farming | 4 |
| Small business | 2 |
| No occupation | 2 |
| **Focus Group Discussions (n = 60)** | |
| **Sex** | |
| Male | 40 |
| Female | 17 |
| **Education Level** | |
| Primary | 23 |
| Secondary | 22 |
| Tertiary | 3 |
| Other | 9 |
| **Occupation** | |
| Religious leader (Christian) | 9 |
| Religious leader (Muslim) | 5 |
| Traditional healer | 14 |
| Farming | 10 |
| Small business | 13 |
| No occupation | 6 |

A caregiver said:

> "For me to be able to know or discover about my son's mental illness, I first of all saw that he was not bathing. Secondly, he was not washing his clothes or caring for his body, and I was like, what is this?" (Male carer, IDI-SA)

Some of the PWLE described their initial experiences with psychosis. Most spoke about hallucinations and events that triggered people to seek care but could not remember how they ended up in the hospital. One PWLE explained:

> "I remember I removed my shirt; people grabbed me, and I started breaking glasses of a certain house, and I found myself in the hospital, and my relatives told me that I was very troublesome, and I was even beating people and throwing stones at them." (Male PWLE, FGD-CZ)

**Table 2.** Summarised main themes and representative quotes.

| Research Questions | Main Theme | Sub-themes | Representative Quotes |
|---|---|---|---|
| 1. How is psychosis experienced and perceived in the community? | Characterization of Psychosis | Description and associated symptoms | *"They do not have time to take care of themselves; they just wander around, and they do not know where they are going and their destination."* (Male PWLE, IDI-CZ) |
| | | Perceived causes of psychosis | *"I think there is a certain worm that disturbs the head. This worm penetrates deep and does its own thing in the head."* (Male traditional healer, FGD-SA) |
| 2. How do community perceptions and experiences relate to treatment-seeking and management, priorities, and preferences for care? | Community treatment-seeking approaches | Influence of aetiological beliefs | *"There is a pervasive belief in the community that mental illness was linked to witchcraft. So, influenced by these beliefs; we attempted traditional treatments."* (Male carer, IDI-SA) |
| | | Perceived treatment effectiveness | *"I saw that my condition was not improving by the time I went to the hospital, so I went to traditional healers, and it was not working either. Several traditional healers tried, but I was not improving."* (Male PWLE, IDI-SA) |
| | | Availability and accessibility of mental health services | *"Sometimes it also depends on the distance of the hospital, how far the hospital is. If the hospital is far, people opt for the traditional healers' because that is what is easily accessible."* (Female carer, IDI-SA) |
| 3. What are the current challenges, and what would represent an ideal system and desirable outcomes among PWLE and their families? | Challenges in managing psychosis | Impacts on daily life | *"For us guardians, we live a very challenging life. You are never productive because you are busy caring for your patient and wondering what will happen next. The patient needs food, but it becomes a challenge for us to go and fetch work for food because of the patient's condition. This is a very burdensome life."* (Female carer, FGD-SA) |
| | | Barriers to treatment | *"We have gone on several occasions only to come back with no medicine. We are poor people; we depend on the government hospitals for medication, and buying is challenging for us."* (Male PWLE, FGD-CZ) |
| 4. What are the current challenges, and what would represent an ideal system and desirable outcomes among PWLE and their families? | Ideal community detection and management system | Active community involvement | *"There is a need to form community groups or committees which will be responsible for mental health issues because one group alone cannot manage. The groups should have 10 members comprising religious leaders, traditional healers, and family representatives."* (Female traditional healer, FGD-SA) |
| | | HSAs engagement | *"They are instrumental because the HSAs are the ones who work in the communities even though they do not come regularly, but that is their job to visit the communities, and they are the ones who can detect mental illness at the community level, and they can offer to advise by telling the family to rush with the patient to the hospital."* (Male carer, FGD-CZ) |
| | | Improved relationship between traditional healers, religious leaders, and healthcare workers | *"I think as leaders, the health workers who work in our community should come and make relationships with us so that when we encounter people with these diseases, we can refer them to the hospital. In addition, we should be trained to know how to report psychosis cases."* (Male religious leader, FGD-SA) |
| | | Availability of resources | *"Increased funding and expanded access to medication would be beneficial. Many individuals with mental illness struggle to afford transportation to hospitals like Zomba Mental Hospital. Making medication available at local clinics would alleviate this burden and ensure continuity of care."* (Male carer, FGD-CZ) |

These descriptions emphasise how psychosis is understood as something which engenders deviations from community norms, and potentially even disrupts these. These deviations are associated with stigma, discrimination, and social isolation.

## Perceived causes

Several causes of psychosis were discussed by participants. The majority regarded psychosis to result from smoking cannabis, excessive alcohol use, and witchcraft. Other causes reported by the participants included physical and emotional abuse, divorce, stress and overthinking, the burden of unpaid loans, fever and headaches, lack of a balanced diet, financial distress, anaemia, HIV, diabetes, rabies, God's punishment for sin, the practising of witchcraft (especially by the elderly), and a tendency to socially withdraw or isolate.

cannabis, known popularly as "*Chamba*" in Chichewa, emerged as a commonly attributed cause among all the stakeholders. In the words of one participant:

*"It happens that a person has started smoking cannabis, a person whose brain is not strong enough to withstand the cannabis. The cannabis affects their brain to the extent that they go mad."* (Male carer, FGD-SA)

Another PWLE narrated:

*"I had lost hope after passing my Malawi school certificate of education with flying colors, but, I was not selected by any college after applying to pursue my education. So, for me to forget about my worries, I started smoking Chamba and drinking beers and ended up developing psychosis in the process. I started doing unnecessary things and my brain wasn't working normally"* (Male PWLE, FGD-SA)

Excessive alcohol usage was also consistently identified as a common cause of psychosis across all the stakeholders. However, unlike carers and PWLE, some traditional healers and religious leaders interpreted heavy drinking and substance use like *Chamba* not only as risky habits for psychosis but also as potential susceptibilities that can be exploited through witchcraft and evil spirits possession. In that way, they can be used to conceal bewitchment as the true cause of psychosis. One traditional healer narrated:

*"Yes, it is possible that the person is an alcoholic, this can lead to psychosis, some people can then bewitch him so that people believe that the cause was alcohol. Others might say the cause is cannabis, but some people also use that as an entry point to bewitch the person so that people may say the person got mad because of the cannabis, when in actual sense it was because he was bewitched."* (Male traditional healer, IDI-CZ)

Biological metaphors and understandings were also evident through narratives of inheritance, with one female participant reflecting:

*"My children can also inherit that same blood of that illness."* (Female carer, IDI-SA)

Another female carer also explained:

*"Some say it is genetic, that someone else in the family suffered from the illness a long time ago"* (Female carer, IDI-CZ)

As expected, given the extant literature [15], psychosis was also commonly attributed to witchcraft. Most of the participants mentioned psychosis was commonly caused by curses and supernatural causes related to witchcraft. Some carers described how others believed that some of their family members bewitched people as part of the rituals to become wealthy, with psychosis emerging as a result. As one carer told us:

*"People ridicule us by saying that we used juju (magic) on the patient; they accuse uncles, aunts, and parents, but I always say, you may think what you may, but this illness comes unexpectedly."* (Female carer, FGD-SA)

Another carer also narrated:

*"In general, people mock us a lot and they say we used juju (magic) on our children to get rich, I mean look at my son, does he look like I used juju on him?"* (Male carer, FGD-SA)

Similarly, some PWLEs also believed they might have been bewitched for their condition to start:

*"For me, I believe that I was bewitched."* (Male PWLE, FGD-CZ)

In some cases, the imagery drawn by the respondents was particularly vivid, highlighting a deep-rooted belief in supernatural attribution; as one traditional healer stated:

*"I think there is a certain worm that disturbs the head. This worm penetrates deep and does its own thing in the head."* (Male traditional healer, FGD-SA)

Financial distress also emerged as an explanation. One religious leader linked psychosis to experiencing financial hardships:

*"Sometimes some of them become mentally ill because of thinking too much as a result of having a lot of financial problems, so because of those challenges, a person thinks too much, such that they develop psychosis in the process."* (Male religious leader, IDI,CZ)

## Community Treatment-Seeking Approaches and care pathways

Most participants reported that traditional healers and religious leaders are often the first points of care for psychosis, frequently preferred over biomedical care. Many believed that traditional healers, using traditional medicine or rituals, were uniquely capable of addressing mental ill-health. Other participants stated that they sought biomedical care only after traditional treatments failed to bring improvement. Conversely, some participants reported turning to traditional healers or religious leaders when biomedical care from hospitals did not yield the desired results of feeling cured. This cycle was driven by three primary factors: local characterizations of mental illness (of the kind mapped in the previous section), perceived treatment effectiveness, and the accessibility of care options.

## Influence of aetiological beliefs

Community, family, and religious understandings around mental illness were reported influential of where the patient would seek help first. As one carer said:

*"There is a pervasive belief in the community that mental illness was linked to witchcraft. So, influenced by these beliefs; we attempted traditional treatments."* (Male carer, IDI-SA)

On the influence of beliefs in witchcraft when seeking treatment, another carer said:

*"If the illness has come because of witchcraft, then even if you go to the hospital for treatment, it cannot work, because it will only need the traditional healer to come in, but, if it has developed as a result of smoking Chamba, then one can get treated at the hospital"* (Female carer, FGD-CZ)

Another carer, in response to a question about where they would first seek help from, reflected:

*"This depends on one's faith, which the household follows or believes in; that was why we decided to visit the prophet first for prayers, and later we decided to go to the hospital."* (Male carer, IDI-CZ)

## Perceived treatment effectiveness

Dissatisfaction with treatment outcomes emerged as a key factor driving individuals to seek alternative treatment services, regardless of whether their initial care was biomedical or non-biomedical. One PWLE said:

*"I saw that my condition was not improving by the time I went to the hospital, so I went to traditional healers, and it was not working either. Several traditional healers tried, but I was not improving."* (Male PWLE, IDI-SA)

Another PWLE elaborated:

*"My condition at that time made them take me to the herbalist, and when they saw that it was not working, they took me to the hospital. That is where they found that it was a mental illness and was started on treatment."* (Female PWLE, IDI-SA)

Some traditional healers and religious leaders also reported that they refer patients to health facilities when their condition does not improve following treatment or prayers. One traditional healer said: *"We have a special relationship with health workers and as I have mentioned, if we know that we will not manage to treat psychosis, we refer the patients to the hospital. We have letters that if we have failed to cure an illness for a week and a half, we should refer them to the hospital"* (Male traditional healer, IDI-SA)
One religious leader also reported:

*"When a person is just screaming, at first we say these are demons. We are convinced that there are demons, people are also convinced and believe that if they go to a sheikh, they will pray for the demons and remove them, or if they go to the pastor, they will pray for the demons to be removed. But, when we see that the condition has not improved, we then advise them to go to the health professionals to diagnose what is wrong with the patient"* (Male religious leader, IDI-CZ)

## Availability and accessibility of mental health services

Apart from local characterisations of mental illness and perceived treatment effectiveness (both linked to social and cultural accessibility), participants also mentioned that the availability and physical accessibility of services influence treatment-seeking approaches. Most participants noted that health facilities are limited and far away from their communities.

As a result, people seek services from traditional healers and religious leaders who are readily available and easily accessible in their communities. One participant said:

*"Sometimes it also depends on the distance of the hospital, how far the hospital is. If the hospital is far, people opt for the traditional healers' because that is what is easily accessible."* (Female carer, IDI-SA)

Another carer also explained:

*"Because we have different beliefs, we went for prayers at a certain man near our home whereby he said it was because of evil spirits that were following him, and bewitched by certain neighbours. So, they prayed for him but it didn't work,hence, we ended up to the health center at chitera village."* (Male carer, FGD-CZ)

However, some traditional healers and religious leaders asserted that cases perceived to have supernatural causes can only be treated by them. On perceived causes of psychosis, one traditional healer explained:

*"Yes, there's a difference. A person who is sick because he has been bewitched cannot get healed at the hospital, that's not possible. We are the ones who can cure them. They stay here, get healed then return to their normal lives, just as they were before"* (Male traditional healer, IDI-SA)

Similarly, on care pathways, one religious leader said:

*"Mostly when seeking help, most of the people go to traditional healers and religious leaders. For example, I was approached to help a patient, I prayed for him and the illness was cured. It helps that the person got better as compared to what he was like before, he was cured. The person returned to normal"* (Male religious leader, IDI-SA)

**Challenges in managing psychosis Impacts on daily life**

Caregivers and PWLE reported that living with psychosis, and caring for individuals with the condition, significantly impacted daily lives. This included carers experiencing physical violence or property damage from those for whom they cared, and a notable lack of community support. On physical violence, one caregiver narrated:

*"When the patients run out of medication, their illness recurs and they beat us up, that is not good hence why we are stressing on the need for medication availability."* (Female carer, FGD-CZ)

Caregivers expressed that providing care for PWLE is an immense responsibility that has disrupted their routines, with much of their daily lives revolving around caregiving tasks. Salient among the challenges that carers experience were that their ability to engage in income-generating activities, such as farming and other essential daily activities, was adversely affected. One carer explained:

*"For us guardians, we live a very challenging life. You are never productive because you are busy caring for your patient and wondering what will happen next. The patient needs food, but it becomes a challenge for us to go and fetch work for food because of the patient's condition. This is a very burdensome life."* (Female carer, FGD-SA)

Similarly, PWLE mentioned that living with psychosis has rendered them unable to obtain an income. This is mainly because they are unable to find jobs due to stigma and discrimination, with other community members described as believing that people experiencing psychosis were incapable of work. One PWLE said:

*"We come across different problems; the first one is that we are discriminated against when it comes to getting piecework in the community; they say we are mad and cannot carry out any task; this happens quite a lot."* (Male PWLE, FGD-SA)

Caregivers also reported experiencing stigma from the community and being suspected of contributing to the patient's condition. One carer mentioned:

*"…but in the community we are often mistreated because of our patients. People claim that we have bewitched them, in order to get rich"* (Female carer, FGD-SA)

**Barriers to biomedical treatment**

Both PWLE and caregivers reported significant challenges in accessing healthcare services. The participants reported difficulties such as the unavailability of psychotropic drugs in health facilities and the long distances to facilities. They also

cited challenges in finding resources to procure medication from private pharmacies and transportation costs to clinics. One PWLE mentioned:

> *"We have gone on several occasions only to come back with no medicine. We are poor people; we depend on the government hospitals for medication, and buying is challenging for us."* (Male PWLE, FGD-CZ)

Some participants also cited unavailability of specialised mental health workers at community clinics as another significant barrier to treatment. One carer said:

> *"In our communities there no special health personnel for those suffering for mental illness and, yet, other illnesses have special health personnel. You will find health personnel for HIV, TB Cancer but none for Psychosis"* (Male carer, FGD-SA)

In some cases, caregivers reported using unconventional methods to treat their patients due to the scarcity of psychotropic drugs at their nearest health facilities. One described:

> *"Okay, like I said, getting help for these young people in our community is not easy, be it from the small hospital or elsewhere, so we use Kachasu [a locally brewed spirit or whiskey] as a calming measure; it helps them. They get better. We use just a half bottle of Kachasu. We do this because of a lack of medication. We know the right medication our patient needs, but the medication is not available in our communities."* (Male carer, FGD-SA)

### Community perspectives on an ideal detection and management system

Based on their diverse experiences, participants were asked to share several recommendations for developing an ideal and sustainable community-based system for psychosis detection and management using prompts from the objectives of PROMISE study. Four approaches were strongly supported: active community involvement, engagement of HSAs, strong partnership between community leaders and health workers, and the availability of resources.

### Active community involvement

The majority of participants recommended forming village or community mental health groups when probed on how their communities could be involved in mental health activities. These groups might address mental health issues arising at the community level, including community sensitisation, assistance in detection and referrals, and close collaboration with the HSAs. The ideal composition of groups would consist of community leaders, caregivers, religious leaders, traditional healers, and community policing members. One female participant said:

> *"There is a need to form community groups or committees which will be responsible for mental health issues because one group alone cannot manage. The groups should have 10 members comprising religious leaders, traditional healers, and family representatives."* (Female traditional healer, FGD-SA)

On awareness and close collaboration between the communities and the HSAs, another participant elaborated:

> *"Awareness campaigns should happen but there should also be committees which will look into these matters, will check how a person is behaving and quickly reach out to the HSAs, this would enable early management"* (Male carer, FGD-CZ)

Another participant emphasized on the need of involving community leaders:

"*The chiefs are the ones that know better what happens in the villages, so after getting the information, they will be able to direct people on what to do, hence I recommend the chiefs to be included in that mental health group*" (Male carer, FGD-SA)

**HSAs engagement**

Most of the participants mentioned that utilising HSAs would be crucial in addressing mental health issues at the community level when probed on significance of HSAs involvement. The participants cited that communities relate well to HSAs because they are based in the communities and are well-known and trusted. This would make it easy for HSAs to screen and encourage families to go to hospitals for clinical care. One carer explained:

"*They are instrumental because the HSAs are the ones who work in the communities even though they do not come regularly, but that is their job to visit the communities, and they are the ones who can detect mental illness at the community level, and they can offer to advise by telling the family to rush with the patient to the hospital.*" (Male carer, FGD-CZ)

Another PWLE added:

"*The HSAs are very important and have a bigger role and capacity to take part in mental health issues, if someone has a mental illness, then it means that the HSA will be needed in that community to refer the person to the main hospital and find ways to help that particular person*" (Male PWLE, FGD-CZ)

Some participants suggested that training the HSAs to provide medication within communities would significantly improve access by reducing transport costs, thereby improving the lives of PWLE and easing the economic burden on their caregivers. One carer said:"*When they are coming near your home and know that there is a patient, they should carry the medication. This will help in the sense that the guardians do not have to travel long distances to the hospital. In my case, we used to go and collect medication at Salima hospital, it was very far for us and very difficult as we do not have transport money. The HSAs are health personnel as well so it shouldn't be difficult for them to bring us the medicines*" (Male carer, FGD-SA)

Although most participants supported the engagement of HSAs, some expressed concerns that their heavy workload could limit their ability to effectively address community mental health needs. One participant expressed:

"*The HSAs have many tasks already, and I believe that they will have a heavy workload by adding mental illness on top of what they already have*" (Male carer, FGD-CZ)

**Improved relationship between traditional healers, religious leaders, and healthcare workers**

Traditional healers and religious leaders mentioned the need for strong partnerships with healthcare workers for an ideal detection and management system. The participants noted that since many patients first seek care from these groups in the communities, they would be crucial in referring families to the hospital. One traditional healer said:

"*we need to work together with the hospital, they must give us bicycles so that we can carry patients to the hospital. We don't want to hold on to patients, what we want is for them to be well…*" (Male traditional healer, IDI-SA)

Most participants acknowledged that proper partnership and collaboration with healthcare workers are currently lacking. One religious leader said:

*"I think as leaders, the health workers who work in our community should come and make partnerships with us so that when we encounter people with these diseases, we can refer them to the hospital. In addition, we should be trained to know how to report psychosis cases."* (Male religious leader, FGD-SA)

## Availability of resources

Despite putting all measures in place, participants mentioned that without vital resources like medication, clinics, transport services, and trained healthcare workers, the system could not function. Many participants cited the availability of psychotropic drugs as the most crucial determinant of a successful psychosis management system. One participant said:

*"..we have talked about medication lengthily, the issue is, whatever discussion we can have about psychosis, it will still come down to medication availability. So, all I can say is that it would be good for these people not to walk long distances to clinics."* (Male carer, FGD-SA)

Another participant highlighted the importance of ensuring the availability of essential medications at the community level to improve access and reduce transportation challenges:

*"Increased funding and expanded access to medication would be beneficial. Many individuals with mental illness struggle to afford transportation to hospitals like Zomba Mental Hospital. Making medication available at local clinics would alleviate this burden and ensure continuity of care."* (Male carer, FGD-CZ)

## Discussion

This study explored key stakeholders' (traditional healers, religious leaders, caregivers and PWLE) experiences and understandings of psychosis and their views on community-based psychosis detection and treatment approaches in Malawi. The findings illuminate the diversity of perspectives around psychosis, and the deeply embedded and entwined social, cultural, and systemic challenges that impact the care of people living with psychosis in Malawi.

Explanations of psychosis were largely attributed to witchcraft and cannabis use (*Chamba*). This aligns with previous research in Malawi, where both witchcraft and *Chamba* are commonly perceived causes of psychosis [10,16]. These findings underscore the complex interplay between cultural beliefs, social contexts, and biological explanations in shaping perceptions of psychosis across diverse community stakeholders. Similar to other studies in Malawi, *Chamba* smoking is often perceived as a form of social deviance, and PWLE suspected of using it are frequently blamed for their conditions, leading to stigma [32]. Beliefs in witchcraft and supernatural causes of mental illness remain widespread in African countries, including Malawi [15]. The popularity of traditional healers and religious leaders as primary points of care for psychosis underscores these characterisations. In the present study, some traditional healers considered supernaturally caused psychosis-like states untreatable by Western medicine. This assertion could partly be attributed to perceived effectiveness of traditional healing. A critical review of studies on the effectiveness of traditional healing in the treatment of mental illness in Africa found it to be potentially effective and widely accepted within communities [33]. This, together with the widespread belief that psychosis is linked to witchcraft, may help explain why some traditional healers and religious leaders might perceive their services as superior to allopathic medicine. Structural factors, such as limited healthcare facilities and medication shortages, reinforced reliance on traditional healers and religious leaders. Additionally, perceptions of treatment effectiveness often shifted pragmatically between traditional, religious, and biomedical services. Similar patterns have been observed in Tanzania's Maasai community, where traditional and Western medicine are used concurrently when initial treatments are deemed ineffective [34]. Similarly, some PWLEs turn to traditional interventions when clinical treatments provide side effects, stigma, or because of structural challenges, such as long distances to healthcare

facilities [8,35]. This complex interplay between culture, religious, and biomedical care pathways is summarised by Van et al. [36], who concluded that patients will continue using traditional medicine and faith healers as long as they perceive them as effective, regardless of the availability of allopathic medicine. However, this argument is disputed by research in rural Ethiopia which has shown that availability of biomedical care through integration of mental health into primary care led to greater uptake of mental health care services, often in preference to traditional or religious healing [37]. In sum, the reflections of our participants in relation to treatment-seeking highlight the influence of – and interactions between - local characterisations of mental illness, perceived treatment effectiveness, and the availability and accessibility of care. They further underscore the often-intertwined roles of biomedical and traditional care systems in addressing mental health conditions.

Caregivers and PWLE reported significant challenges, including disruptions to daily life, economic strain, stigma, and barriers to healthcare access, such as limited mental health services and shortages of psychotropic drugs. These findings are consistent with evidence from other LMICs [23, 38,39]. In response to these adversities, certain practices - such as alcohol use, neglect and verbal threats - can be detrimental to the well-being of individuals with mental illness [40]. In the present study, only one clear example of a potentially harmful practice was identified, wherein a participant was given alcohol to subdue them. Overall, it become clear from our dataset that economic, material, and social precarity intersected to deepen the challenges for PWLE and those who cared for them. Caregivers and PWLE need better support systems. Our findings emphasize the need for community-led interventions to enhance health outcomes and daily functioning. Evidence from several African countries has shown that lived experience-led mental health initiatives can improve treatment adherence, rights advocacy and social reintegration of PWLE [27,41]. Caregivers could benefit from psychoeducation, peer support networks, and financial assistance programs to reduce the caregiving burden.

Finally, the formation of community mental health groups was endorsed as a strategy to raise awareness and facilitate the referral of untreated patients. This aligns with expert insights suggesting that untreated psychosis detection efforts require active community involvement across all stakeholders [42]. Another key recommendation was the engagement of HSAs, which aligns with the objectives of the PROMISE study, aimed at training HSAs in community-based psychosis detection and management in Malawi [5]. Additionally, integrating traditional healers and religious leaders into mental health referral systems could improve early identification and reduce DUP. The traditional healers and religious leaders in this study expressed willingness to collaborate with mental health workers to establish community-focused referral systems that could significantly improve early detection. A large West African clinical trial demonstrated that collaborative mental health models with traditional healers are feasible and cost-effective [43]. The availability of human and clinical resources, particularly antipsychotic medications, was also reported as pivotal for sustaining community-based psychosis management. Like Malawi, many LMICs face challenges in psychotropic drug supply chain management [44]. Ensuring consistent medication availability at district and primary healthcare levels is vital to strengthening community mental health services.

## Strengths and limitations

The study's main strength was the diverse representation of the community stakeholders, recruited from different geographical communities to capture broader perspectives on community perceptions, treatment practices, and acceptable pathways for psychosis management in Malawi. This inclusivity enhances the study's relevance for informing community-based mental health interventions.

However, several limitations should be acknowledged. First, the FGDs, traditional healers and religious leaders were grouped together. Similarly, caregivers and PWLE were also grouped together. This grouping may have influenced participants' willingness to express their views openly due to potential social hierarchies, perceived judgement, or fear of reprimand. To mitigate this, the research assistants emphasised confidentiality and encouraged respectful dialogue during

discussions. Further, while purposive sampling ensures rich and diverse insights, it may have introduced selection bias, limiting the breadth of perspectives. Moreover, the findings reflect participants' experiences in the Salima and Chiradzulu districts and may not be generalisable to all Malawi communities despite consistency with findings from similar low-resource settings in sub-Saharan Africa [6,7]. Additionally, the study was undertaken in the context of the wider PROMISE study and data contributed to intervention design. This shaped what was asked in the focus groups and inter-views, thus providing a wider context for the data collection and accountability to communities in terms of a pathway to potentially changing mental health services. However, this also potentially limited the scope of the study and methods. Despite these limitations, the study provides valuable insights into community perceptions and practices of treatment-seeking for psychosis, offering a strong foundation for developing future mental health interventions.

## Conclusion

This study underscores the critical role of cultural norms and practices, alongside social and economic structures and interpersonal challenges, in shaping perspectives on psychosis and its treatment in Malawi. Community knowledge and understanding of psychosis remain limited, continuing to influence treatment-seeking behaviors and contributing to stigma and discrimination toward PWLE and their caregivers. This underscores the need for awareness initiatives to promote early detection and facilitate timely linkage to care. Furthermore, the present study indicates that traditional and religious leaders remain influential in care provision, underscoring the importance of their engagement in psychosis care in Malawi.

## Recommendations

Addressing psychosis detection and management challenges at a community level requires an integrated approach that (1) engages HSAs in early psychosis detection, (2) strengthens collaboration between traditional healers and healthcare workers through structured referral pathways, (3) improves access to psychotropic medication at primary health facilities, and (4) enhances caregiver and PWLE involvement in mental health support networks. Future interventions should priori-tise task-sharing, sustainable medication supply chains, and culturally sensitive community outreach programs to improve psychosis management in Malawi.

## Supporting information

**S1 Checklist. COREQ.**
(DOCX)

**S2 Checklist. Inclusivity in global research.**
(DOCX)

## Acknowledgments

We thank and appreciate all the Chiradzulu and Salima participants for their time in this study. We also acknowledge the representative from the Chiradzulu district mental health team (Mr. Nyangulu) and the Salima MeHUCA representative (Ms. Shineva) for their assistance with participant recruitment and study logistics.

## Author contributions

**Conceptualization:** Dennis Chasweka, Demoubly Kokota, Charlotte Hanlon, Stephen M. Lawrie, Lucinda Manda-Taylor, Sumeet Jain.

**Data curation:** Dennis Chasweka, Kate Chidzalo, Wakumanya Sibande, Martyn Pickersgill.

**Formal analysis:** Dennis Chasweka, Wakumanya Sibande, Martyn Pickersgill, Charlotte Hanlon, Stephen M. Lawrie, Lucinda Manda-Taylor.

**Funding acquisition:** Charlotte Hanlon, Stephen M. Lawrie.

**Investigation:** Dennis Chasweka, Kate Chidzalo, Michael Udedi, Charlotte Hanlon, Stephen M. Lawrie.

**Methodology:** Dennis Chasweka, Demoubly Kokota, Stephen M. Lawrie, Lucinda Manda-Taylor, Sumeet Jain.

**Project administration:** Dennis Chasweka, Wakumanya Sibande.

**Resources:** Demoubly Kokota.

**Supervision:** Stephen M. Lawrie, Lucinda Manda-Taylor, Sumeet Jain.

**Writing – original draft:** Dennis Chasweka.

**Writing – review & editing:** Kate Chidzalo, Wakumanya Sibande, Demoubly Kokota, Michael Udedi, Martyn Pickersgill, Charlotte Hanlon, Stephen M. Lawrie, Lucinda Manda-Taylor, Sumeet Jain.

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
