## [Decision Letter · Decision Letter 0]

7 Jan 2026

PMEN-D-25-00520

“I think there is a certain worm that disturbs the head”: Perspectives from traditional healers, religious leaders, caregivers, and people with lived experience of psychosis in Malawi.

PLOS Mental Health

Dear Dr. Chasweka,

Thank you for submitting your manuscript to PLOS Mental Health. After careful consideration, we feel that it has merit but does not fully meet PLOS Mental Health’s publication criteria as it currently stands. Therefore, we invite you to submit a revised version of the manuscript that addresses the points raised during the review process.

We look forward to receiving your revised manuscript.

Kind regards,

Vitalii Klymchuk, Ph.D., D.Sc.

Academic Editor

PLOS Mental Health

Journal Requirements:

1. Please ensure that you refer to Table 1 in your text as, if accepted, production will need this reference to link the reader to the table.

Additional Editor Comments (if provided):

Reviewers' comments:

Reviewer's Responses to Questions

Comments to the Author

1. Does this manuscript meet PLOS Mental Health’s publication criteria? Is the manuscript technically sound, and do the data support the conclusions? The manuscript must describe methodologically and ethically rigorous research with conclusions that are appropriately drawn based on the data presented.

Reviewer #1: Partly

Reviewer #2: Yes

2. Has the statistical analysis been performed appropriately and rigorously?

Reviewer #1: No

Reviewer #2: Yes

3. Have the authors made all data underlying the findings in their manuscript fully available (please refer to the Data Availability Statement at the start of the manuscript PDF file)?

Reviewer #1: No

Reviewer #2: Yes

4. Is the manuscript presented in an intelligible fashion and written in standard English?

Reviewer #1: No

Reviewer #2: Yes

5. Review Comments to the Author

Reviewer #1: Topic: Your topic, ""I think there is a certain worm that disturbs the head”: Perspectives from traditional healers, religious leaders, caregivers, and people with lived experience of psychosis in Malawi" is not precise. In its present state, it looks a "sweeping statement", trying to generalize or include everything. Think critically and make a more precise title.

Introduction: Again, the introduction is less focus because of the topic. The most confusing thing emanating from the topic is "Perspectives from traditional healers, religious leaders, caregivers, and people with lived experience of psychosis in Malawi". Perspective on what??? Just perspective makes it lose and less focus, thus affecting everything in your study, including result section.

The concluding part of your introduction, showing the research questions is very good. Use this to guide the structuring of your result section.

Result:

-First present a table summarizing the themes and sub-themes before you start using them in the findings. Do not remove table 1 but add another table of themes and sub-themes below it.

-Remove all discussions from result section. In result section, present only what you found in relation to your study objectives.

-The concluding part of your introduction section, showing the research questions is very good. Use this to guide the structuring of your result section. In its present form, the structure of the result section is not tailored to the research questions or study objectives, making it difficult and boring to read.

Discussion: The discussion is also not clear because of the way results are presented. Once you have addressed the concerns in the result section, you can also aggressively revise the discussion section

Reviewer #2: This manuscript presents perspectives of community stakeholders on psychosis in Malawi. The topic is highly relevant to global mental health, especially in low- and middle-income countries where various treatment options are common and understudied.

The study is methodologically rigorous. The sampling strategy is appropriate for the exploratory qualitative design, data collection procedures are clearly described, and the analytic process demonstrates transparency, reflexivity, and credibility. The inclusion of multiple stakeholder groups—traditional healers, religious leaders, caregivers, and people with lived experience—adds significant depth and triangulation to the findings. The results are clear, supported by strong quotes, and organized by themes. The discussion situates the findings within the context of existing research and connects community beliefs, health system challenges, and potential interventions. The conclusions are based on the data and stay within the study’s scope. Ethical concerns are adequately addressed, and the Data Availability Statement accurately explains restrictions on participant confidentiality. However, the manuscript needs minor revision on clarity, data saturation justification, generalizability and context, minor grammatical issues, and consistency. My detailed comments are below.

1. Clarity on Ethical Approval Timeline

The cover letter explains the confusion regarding ethics approval dates, but the manuscript itself should briefly clarify in the Methods or Ethics Statement that approval was granted prior to data collection (July 2023 for October–December 2023 enrollment).

2. Data Saturation Justification

The authors reference Hennink et al. to justify sample size but could briefly elaborate on how saturation was assessed in practice (e.g., no new themes emerged in later interviews/FGDs).

3. Generalizability and Context

The Limitations section appropriately notes that findings may not be generalizable to all of Malawi. Consider briefly discussing how findings might resonate with (or differ from) other rural, low-resource settings in sub-Saharan Africa.

4. Minor Writing Edits:

Check for minor grammatical issues (e.g., “No any competing interests” → “The authors declare no competing interests”).

In conclusion. Cultural norms… → should be one sentence, not split.

Ensure consistency in abbreviations (e.g., PWLE is defined but sometimes written as PWLEs (e.g., 146 line number)) and “Indian hemp” vs “cannabis” (both are fine, but consistency matters).

6. PLOS authors have the option to publish the peer review history of their article (what does this mean?). If published, this will include your full peer review and any attached files.

Do you want your identity to be public for this peer review? For information about this choice, including consent withdrawal, please see our Privacy Policy.

Reviewer #1: No

Reviewer #2: No

Figure Resubmissions:

---

## [Editor Report · Decision Letter 1]

20 Apr 2026

"I think there is a certain worm that disturbs the head”: Community perspectives and care pathways for psychosis in Malawi

PMEN-D-25-00520R1

Dear Mr Chasweka,

We are pleased to inform you that your manuscript '"I think there is a certain worm that disturbs the head”: Community perspectives and care pathways for psychosis in Malawi' has been provisionally accepted for publication in PLOS Mental Health.

Best regards,

Vitalii Klymchuk, Ph.D., D.Sc.

Academic Editor

PLOS Mental Health